# Co-crystallisation and humanisation of an anti-HER2 single-domain antibody as a theranostic tool

Kovilen Sawmynaden[1]*, Nicholas Wong[2], Sarah Davies[1], Richard Cowan[3], Richard Brown[1], David Tang[1], Maud Henry[1], David Tickle[1], David Matthews[1], Mark Carr[3], Preeti Bakrania[1], Hong Hoi Ting[2], Gareth Hall[3]*

1 LifeArc, Open Innovation Campus, Stevenage, United Kingdom, 2 NanoMab, Shanghai, China, 3 Department of Molecular and Cell Biology, Leicester Institute of Structural and Chemical Biology, University of Leicester, Leicester, United Kingdom

* gh126@leicester.ac.uk (GH); kovilen.sawmynaden@lifearc.org (KS)

## Abstract

Human epidermal growth factor receptor-2 (HER2) is a well-recognised biomarker associated with 25% of breast cancers. In most cases, early detection and/or treatment correlates with an increased chance of survival. This study, has identified and characterised a highly specific anti-HER2 single-domain antibody (sdAb), NM-02, as a potential theranostic tool. Complete structural description by X-ray crystallography has revealed a non-overlapping epitope with current anti-HER2 antibodies. To reduce the immunogenicity risk, NM-02 underwent a humanisation process and retained wild type-like binding properties. To further de-risk the progression towards chemistry, manufacturing and control (CMC) we performed full developability profiling revealing favourable thermal and physical biochemical 'drug-like' properties. Finally, the application of the lead humanised NM-02 candidate (variant K) for HER2-specific imaging purposes was demonstrated using breast cancer HER2+/BT474 xenograft mice.

## Introduction

There are over 2 million diagnoses of breast cancer per year, with a 70–80% survival rate if detected early and prior to organ metastases [1, 2]. Human epidermal growth factor receptor-2 (HER2) is a well characterised cell-surface glycoprotein (~ 185 kDa), first identified as the product of an oncogene in the late 1980s. Overexpression of HER2 has been observed in many cancers, most notably breast cancer, wherein 30% of breast cancers exhibted a two to twenty-fold increase in gene amplification, correlating with higher recurrence rates and shorter overall survival times for patients [3, 4]. From population-based studies in the US, HER2-positive breast cancer prevalence is estimated to be between 15 and 19% [5]. Overexpression of the HER2 receptor in healthy cells is able to initiate 'cancer-like' behaviour, including increased cell division and tumour formation [6, 7]. Correspondingly, there is strong evidence that suggests that HER2-positive (HER2+) breast cancers are more aggressive than HER2-negative (HER2-) breast cancers, leading to poorer patient prognosis and unfavourable tumour

**Data Availability Statement:** Data are deposited with the World Wide Protein Data Bank, https://www.ebi.ac.uk/pdbe/entry/pdb/7qvk [DOI: 10.2210/pdb7qvk/pdb].

**Funding:** I confirm that I have mentioned all organizations that funded my research in the Acknowledgements section of my submission, including grant numbers where appropriate." a) This work was funded by the research charity LifeArc, under contractual arrangement with NanoMab. LifeArc funded the research activity carried out by the University of Leicester group. b) LifeArc and NanoMab were actively involved in the study design, data collection and analysis, as well the decision to publish and preparation of the manuscript. c) Kovilen Sawmynaden, Sarah Davies, Richard Brown, David Tang, Maud Henry, David Tickle, David Matthews and Preeti Bakrania received a salary from LifeArc. Nicholas Wong and Hong Hoi Ting received a salary from NanoMab.

**Competing interests:** The authors declare the following financial interests/personal relationships which may be considered as potential competing interests: LifeArc provided the antibody humanisation service(s) under contractual arrangement with NanoMab. This does not alter our adherence to PLOS ONE policies on sharing data and materials.

characteristics [3, 8, 9]. The extracellular domain (ECD) of HER2 is the target of current monoclonal antibody (mAb) treatments for HER2+ breast cancer, including Trastuzumab (Herceptin®) and Pertuzumab (Perjeta®). In 2019, an antibody-drug conjugate, Trastuzumab Emtansine (Kadcyla®) was approved for adjuvant treatment of early-stage breast cancer [10]. Hence, the continual development of antibody-based modalities has provided new opportunities to both treat and diagnose HER2+ tumours.

Single-domain antibodies (sdAbs or VHHs) can be isolated from the ancestral, heavy-chain-only antibodies of camelids [11]. Such modalities have been shown to exhibit robust biophysical properties and can often retain antigen-specific binding activity following chemical or thermal stress [12]. Additionally, their small ~ 15 kDa size, and often extended third complimentary determining region (CDR), has made them particularly attractive as diagnostic 'tools'. This is partially due to their potential to sample surfaces traditionally inaccessible to larger monoclonal antibodies [13, 14]. Combining diagnostic tools with radioisotopes further opens the possibility of a 'two-in-one' approach. This is particularly pertinent to the field of 'Theranostics', whereby a single antibody is utilised to both stratify patient groups and subsequently deliver treatment [15]. In the case of sdAbs, these can be simply appended with chemically-defined linkers and 'loaded' with nuclides that are either appropriate for diagnostic-based imaging (*e.g.* technetium-99m) or cancer cell killing/therapy (*e.g.* actinium-225 or iodine 131) [14]. Such a strategy allows the swapping 'in or out' of radiolabels, according to patient need, and results in a true personalised medicine experience.

Although sdAbs fall within the range for renal excretion ($< 69$ kDa) [16], repeated administration of camelid-derived biologics with low similarity to human germline ($< 75\%$), has the potential to induce an increased risk of immunogenicity. Overcoming this problem is well recognised for therapeutic monoclonal antibodies (mAbs), especially those of animal origin, and is generally mitigated by antibody engineering processes termed 'humanisation' [17, 18]. Hence, it is sensible to adopt analogous approaches for sdAbs to also minimise the risk of immunogenicity, especially where repeated administration or dosing is required, as may be the case for therapeutic or theranostic purposes. In this study, a highly specific anti-HER2 sdAb candidate has been identified as having a non-overlapping epitope to current frontline treatment options. Following complete epitope-paratope description, obtained by co-crystallisation with human epidermal growth factor receptor-2-extracellular domain HER2-ECD, the lead candidate (NM-02) was subject to humanisation processes and full developability profiling in order to further de-risk progression towards chemistry, manufacturing and control (CMC). Lastly, the lead humanised sdAb has been demonstrated as a HER2-specific imaging tool *in vivo*, within a human breast cancer xenograft mouse model.

## Materials and methods

### Immunisation and phage library construction

Recombinant HER2-ECD (produced in HEK cells and $\geq 90\%$ pure via sodium dodecyl sulphate polyacrylamide gel electrophoresis (SDS-PAGE)) was used to immunise two domestic camels (*Camelus bactrianus*) over a period of several weeks. Peripheral blood lymphocytes (PBL) were harvested, ribonucleic acid (RNA) extracted and VHH genes amplified to construct a phage display library. Incorporation of relevant sized inserts was confirmed using polymerase cain reaction (PCR) and diversity determined by measuring colony-forming units (CFU).

## Cross-reactivity testing of NM-02 to recombinant HER2-ECDs

ELISA microplates were coated with human HER2-ECD-Fc or mouse HER2-ECD-Fc (R&D systems, Minneapolis, MN) and blocked with 1% bovine serum albumin (BSA) at room temperature. Single-domain antibody NM-02 was serially diluted (from 160 nM) and added to the microplate, before binding was probed using a mouse anti-HA secondary antibody (Biolegend, San Diego, CA, USA). An anti-mouse IgG-alkaline phosphatase (Proteintech, Rosemont, IL, USA) in the presence of TMB (eBioscience, San Diego, CA, USA) was used for colorimetric detection. Absorbance at 450 nm was recorded on a microplate reader (Bio-Tek, Agilent, USA).

## Cell-surface binding of NM-02 to cancer cell lines

Adherent cancer cell lines (BT474, SKOV3 and MDA-MB-231) were detached from flasks using a commercially-sourced, non-enzymatic cell dissociation buffer, containing a proprietary mixture of chelators (Cell Stripper®; Corning, NY, USA) and re-suspended in phosphate buffered saline, pH 7.4 (PBS) ($2x10^6$ cells/mL). Cells were first incubated with FcR blocking reagent (Miltenyi Biotec, Germany) followed by NM-02 and Alexa Fluor 488 anti-HA antibody (Biolegend, San Diego, CA, USA). All incubations were performed at 4 ºC in the dark and cells were washed twice with PBS between each step. Flow cytometry analyses were performed on a FACSCalibur™ flow cytometer (BD Biosciences, UK).

## Competition of NM-02 with Trastuzumab and Pertuzumab

Human HER2 ECD-Fc was coupled to a CM5 sensor chip (800 resonance units) pre-loaded into Biacore® 3000 system (Cytiva). NM-02, Trastuzumab (Roche) or Pertuzumab (Roche) were sequentially injected either alone or in combination at 30 μL/min in 10 mM HEPES, pH 7.4, 150 mM NaCl, 3 mM EDTA and 0.05% P20 (HBS-EP) buffer in two 180 sec binding phases. The curves were analysed using BIAevaluation (version 4.1) and visually interpreted.

## Microarray screening

Microarray library screening was performed by Retrogenix (www.retrogenix.com) and comprised fixed HEK293 cells overexpressing 5484 full-length plasma membrane and membrane-tethered secretory proteins. Recombinant NM-02 (2 μg/mL) was either HA- and/or His-tagged and 'hits' detected using anti-tag secondary antibodies conjugated to AlexaFluor 647. For additional assay details *see* Freeth and Soden [19].

## Production of recombinant humanised single-domain antibodies

DNA inserts were codon-optimised (*Cricetulus griseus*), synthesised and subcloned into pcDNA3.1 (+) by GenScript. A human Ig-*k* (IGKV3D-15) leader sequence was incorporated upstream of the coding sequences, for secretory expression. A His-tag (including a short GSG linker) was incorporated at the C-terminus to facilitate protein purification.

Prepared DNA plasmids were transiently transfected into ExpiCHO-S™ cells and cultured according to the manufacturer's instructions (Thermo Fisher Scientific). Supernatants were harvested by centrifugation, filter-sterilised (Stericup®; Merck) and supplemented with 25 mM HEPES (pH 7.2) and 0.05% sodium azide. Harvested supernatants were stored at 4˚C. Single-domain antibodies were purified from the supernatant by affinity chromatography (HisTrap™ Excel column; Cytiva) and size-exclusion (HiLoad (16/600) Superdex 75; Cytiva) chromatography (SEC), into a final buffer of PBS (10 mM sodium phosphate, pH 7.5, and 150 mM NaCl).

Successful removal of large sample aggregate(s) was confirmed using dynamic light scattering (DLS). Data was acquired using a Zetasizer APS (Malvern Panalytical, UK) from purified samples (1–3 mg/mL) to determine the hydrodynamic radius (Z-Average diameter) and polydispersity index (PDI). All measurements were performed at room temperature.

Endotoxin measurements were performed using a Endosafe$^®$ nexgen-PTS™ device (Charles River, USA).

## Production of recombinant HER2-ECD for crystallisation

Expression and purification of HER2-ECD (23–646; UniProt ID P04626) was performed by Proteros (www.proteros.com). The protein was expressed in HEK293F cells and purified by a multi-step chromatographic procedure, including affinity capture *via* Ni-NTA, deglycosylation and size-exclusion chromatography. Lastly, the sample was concentrated to 18 mg/mL, flash-frozen in liquid nitrogen and stored at minus 80 ˚C. The storage buffer comprised: 20 mM HEPES, pH 7.0, and 150 mM NaCl.

## Crystallisation of NM-02 bound to HER2-ECD

A protein complex, comprising NM-02 and HER2-ECD was assembled by mixing the sdAb and target ligand at a molar ratio of 2:1. The mixture was incubated for 45 minutes at 4˚C, concentrated using a centrifugal concentrator (Vivaspin-20; Sartorius) to a final volume of 2 mL and purified by size-exclusion chromatography (Superdex 200 10/300 column; Cytiva). The mobile phase comprised 20 mM Tris-HCl, pH 7.5, and 50 mM NaCl. Fractions corresponding to the protein complex peak were pooled and re-concentrated (Vivaspin-20; Sartorius) to 4 mg/mL.

Initial crystallisation trials were performed using sitting drop vapor diffusion and 96-well block screens (Molecular Dimensions) at 20 ˚C. Optimal crystals grew in 13% PEG6000, 0.1 M MES (pH 6.5) and 7% 2-methyl-2,4-pentanediol. A cryoprotectant consisting of 20% 2-methyl-2,4-pentanediol (v/v) in mother liquor was used. X-ray diffraction data were collected at the Diamond Light Source (Oxford, UK) using beamline I24.

Diffraction data were indexed and integrated in XDS [20] and scaled to 3.1 Å using AIMLESS, as part of CCP4 [21]. A Matthews Coefficient [22] calculated one molecule of NM-02 and one molecule of HER2-ECD present in the asymmetric unit, giving a solvent content of 67%. A homology model of NM-02 was generated with SWISS-MODEL [23]. The structure of the NM-02 in complex with HER2-ECD was determined by molecular replacement, using the program PHASER [24] input with the structures of HER2-ECD (PDB ID 5MY6) and the NM-02 homology model used in the search. Model building and structural refinement was carried out with Coot [25] and REFMAC 5.0 [26] using restrained refinement and isotropic B-factors. X-ray data collection and refinement statistics are given in S1 Table.

## Binding kinetics of humanised single-domain antibody variants to HER2-ECD

HER2-ECD biotinylated at single site (Sino Biological) was applied to a streptavidin (SA) sensor chip (0.1 µg/mL) pre-loaded into a Biacore 8 K instrument (Cytiva). The flow rate and temperature were set to 100 µL/min and 25˚C, respectively. All experiments were performed in 10 mM HEPES, pH 7.4, 150 mM NaCl, 3 mM EDTA and 0.05% P20 (HBS-EP+) buffer. Single-domain antibodies were injected in a multi-cycle kinetics method using a 2-fold dilution series starting at a top concentration of 5 nM, including replicates (Rmax = 15–20 RU). Data was processed using Biacore Insight software and fitted to a 1:1 model.

## Melting temperature (Tm) measurements

Single-domain antibody samples (1 mg/mL) were loaded (in triplicate) into an Uncle instrument (Unchained Labs, USA). Samples were thermally ramped (0.5˚C/min) and transiently held (120 sec) at discrete temperature points between 25˚C and 95˚C. A complete fluorescence spectrum (250–720 nm) was captured for each sample at every temperature point and changes monitored using the Barycentric mean (BCM). Melting temperatures were determined by identifying the midpoint of the fluorescence transition (dy/dx = 0). Static light scattering (SLS) was measured at 266 nm and 473 nm.

## Freeze-thaw stress-testing

Single-domain antibody samples were concentrated to 7.5 mg/mL and subject to ten freeze-thaw cycles by placing at minus 80˚C (15 mins) followed by room temperature thawing (15 mins). Aggregation content was subsequently determined using an analytical SEC column (AdvanceBio SEC 130Å) fitted to a 1260 Infinity High-performance liquid chromatography (HPLC) system (Agilent) and coupled in series to a multi-angle static light scattering (MALS) detector and differential refractometer (Wyatt Technology). All samples were injected at a flow rate of 0.4 mL/min into a mobile phase comprising PBS at room temperature. The refractive index increment (dn/dc) on the MALS detector was set to 0.185 for protein analysis. Data was analysed using ASTRA (version 7; Wyatt Technology).

## Serum stability

Single-domain antibody samples (His-tagged) were added in duplicate to a 96-well plate, containing either human, mouse or cynomolgus sera (supplied by BioIVT) and incubated for 7 days at 37˚C (final concentration = 100 μg/mL). Retention of binding activity was determined by ELISA versus single-domain antibody samples incubated in PBS, under the same conditions. The ELISA set-up comprised recombinant human HER2-Fc (1 μg/mL) coated to the surface of a MaxiSorp™ 384 well plate (Nunc). Sera were serially diluted, applied and binding probed using an anti-HIS-HRP secondary antibody conjugate (in the presence of TMB substrate). Absorbance was measured at 650 nm and data fit using non-linear regression to obtain half-maximal effective concentrations ($EC_{50}$).

## Cross-interaction chromatography (CIC)

A (1 mL) HiTrap NHS-activated column (Cytiva) was loaded with $\geq$ 10 mg human polyclonal IgG as a coupling ligand. A control column was prepared in the same way, but without ligand loading. Single-domain antibody samples (0.5 mg/mL) were injected separately onto each column and a retention factor ($k$') determined according to:

$$k' = [(Tr - Tm)]/Tm$$

where $Tr$ is the retention time of the sample on the polyclonal IgG column and $Tm$ is the retention time on the control column. Data was acquired on a 1260 Infinity HPLC system (Agilent) using a mobile phase comprising PBS (+ 0.01% $NaN_3$) and a constant flow-rate (0.1 mL/min.).

## SPECT/CT imaging and HER2 positive BT474 xenograft mouse model

Six-week-old, female Balb/C nu/nu mice were purchased from Charles River. All animals were acclimated for a minimum of 7 days. During acclimation amd study periods the animals were socially housed in individually ventilated cages with bedding in an environmentally monitored

room. The animal holding rooms were set to maintain a temperature of 22°C ± 4°C and a target relative humidity of 50% ± 20%. Animal rooms were maintained on 12 hour alternating light and dark cycles. Mice were subcutaneously inoculated on the right flank with BT474 cells (6 x10$^6$ cells suspended in 1:1 (v/v) matrigel:PBS) following implantation of estrogen pellets, using buprenorphine analgesic. The animals were observed daily for any abnormalities indicative of health problems, to prevent suffering. Only healthy animals were placed on the study. Once on study, unique identifiers were marked on animals by permanent marker, ear punch or tattoo. Sterile food and sterilised tap water were made available *ad libitum* throughout acclimation and the biological phase of the study. Tumour growth and animal body weight were monitored twice a week. Tumour size was calculated according to:

$$\text{Tumour size} = 0.52*\text{length}*\text{width}^2$$

Approximately three to four weeks after inoculation, when the average xenograft size had reached 200–300 mm$^3$, mice were randomly assigned to four groups for *in vivo* imaging. Mice in group A (n = 5) were intravenously administered 15.3 ± 2.6 MBq of [$^{99m}$Tc]-NM-02 (equivalent to 20 μg of radiolabelled NM-02). Mice in group B (n = 5) were also intravenously administered 16.3 ± 2.6 MBq of [$^{99m}$Tc]-NM-02, plus an additional 180 μg of unlabelled (or 'cold') NM-02. Mice in group C (n = 5) were intravenously administered 19.4 ± 0.5 MBq of [$^{99m}$Tc]-variant K (equivalent to 20 μg of radiolabelled humanised variant K). Mice in group D (n = 5) were also intravenously administered 23.5 ± 1.0 MBq of [$^{99m}$Tc]-variant K, plus an additional 180 μg of unlabelled (or 'cold') humanised variant K. Whole-body SPECT/CT images were acquired 1.5 hr post-injection in a VECTor6-CTXUHR (MILabs) preclinical imaging system under isoflurane anaesthesia. Biodistribution of [$^{99m}$Tc]-NM-02 and variant K was calculated based on SPECT/CT images. Biodistribution at 1.5 hr post-injection for different groups was compared by two-sample unequal variance t-tests. Mice under anaesthesia were sacrificed by cervical dislocation, after confirmation that all SPECT/CT images were technically acceptable. All animal studies were performed by Invicro LLC, operating at the premises of Queen Mary University of London, London, UK. The study was ethically reviewed and carried out in accordance with the UK Animals (Scientific Procedures) Act 1986, UK Home Office regulations governing animal experimentation and in accordance with the EU Directive 2010/63/EU. The techniques (regulated procedures) performed were approved for the purposes of this study by the Queen Mary University of London Project Assessment Committee.

## Results

### Discovery and characterisation of anti-HER2 candidate: NM-02

An initial total of 40 anti-HER2 sdAbs were identified from panning an assembled phage display library (library diversity = 2 x 10$^9$ colony-forming units) and comprised 19 families, based upon the amino acid sequences of CDR 1, 2 and 3. A representative subset were then triaged for affinity to target, adherence to cancer cell lines, competition with available therapeutics and off-target specificity resulting in the identification of NanoMab-02 (NM-02) as the lead candidate. Using an enzyme-linked immunosorbent assay (ELISA) candidate NM-02 demonstrated both high affinity (EC$_{50}$ = 1.2 nM) and specificity towards human HER2-ECD, with no detectable cross-reactivity to murine HER2-ECD (Fig 1a). The ability of NM-02 to bind human HER2 on ovarian and breast cancer cell lines was further evaluated using flow cytometry. A clear shift in fluorescence intensity was observed for HER2+ cells (BT474 and SKOV3) versus the negative control cell line (MDA-MB-231), confirming that NM-02 was able to recognise cell-surface expressed HER2 from different cancer types (Fig 1b). Competition experiments with existing HER2 therapies (Trastuzumab and Pertuzumab) were performed using

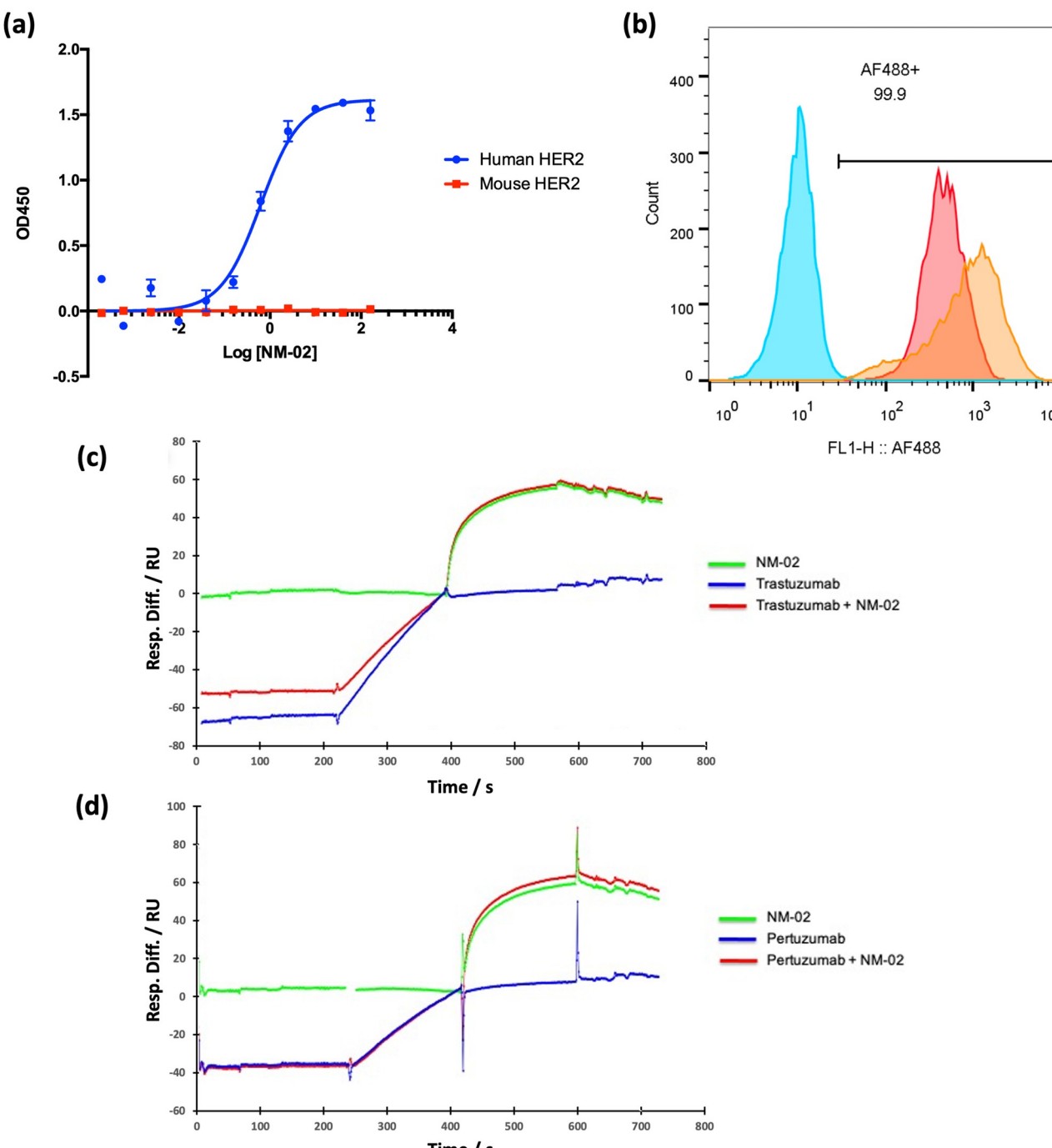

**Fig 1. Biochemical characterisation of anti-HER2 sdAb, NM-02.** (a) Cross-reactivity testing of NM-02, demonstrating specific binding to recombinant human HER2-ECD (blue) but not recombinant murine HER2-ECD (red). (b) Cell surface adherence of NM-02 to HER2+ (BT474-Red and SKOV3-Orange) but not HER2- (MDA-MB-231-Blue) cell lines using flow cytometry. (c) and (d) Competition testing by surface plasmon resonance, showing that NM-02 (green) binds HER2-ECD following the sequential binding of either Trastuzumab (c) or Pertuzumab (d). Therefore, the NM-02 epitope is non-competitive with frontline therapeutics.

surface plasmon resonance (SPR). A desirable, non-overlapping epitope for NM-02 (Fig 1c and 1d) was identified, implying that the sdAb could be utilised alongside existing treatment regimes, without steric interference.

## Off-target microarray screening

To mitigate any potential risk associated with off-target toxicity, NM-02 was subjected to an extensive *in vitro* microarray accessed under commercial agreement with Retrogenix™. Here, a dual tagged version of NM-02 (comprising a hexahistidine and hemagglutinin (HA) epitope tag) was used to probe a library comprising 5484 full-length human plasma membrane proteins and cell surface-tethered, human secreted proteins. After filtering non-reproducible and/ or non-specific hits, a single hit to HER2 soform 1 remained (Fig 2). Therefore, it was determined that NM-02 exhibited excellent specificity for HER2-ECD and confirmed its tractability as a drug discovery lead.

## Structure of NM-02 in complex with HER2-ECD

To facilitate single domain antibody engineering and epitope-paratope relationship, the structure of NM-02 in complex with HER2-ECD was solved by X-ray crystallography (PDB: 7QVK). As the epitope was initially undefined, the largest crystallisable construct of

**Fig 2. Off-target microarray screening of anti-HER2 sdAb, NM-02.** A specific interaction between duel-tagged NM-02 and ERBB2/HER2 (green) was observed using both anti-His (a) and anti-HA (b) monoclonal antibodies labelled with AlexaFluor 647. Non-specific spotting was defined as being detected in both the NM-02 and control (PBS) samples. Two replicates (Rep) shown per sample. A key for the spotting pattern is also shown inset.

HER2-ECD, comprising residues 23–646 (UniProt ID: P04626) was pursued. Crystals of the purified complex were obtained and diffracted to 3.1 Å (S1 Table). Analysis of the diffraction data indicated that the asymmetric unit contained one molecule of NM-02 in complex with one molecule HER2-ECD.

Interpretable electron density was obtained for HER2-ECD residues Gln24 to Val541 and NM-02 residues Gln3 to Ser123 (numbering residues sequentially from N- to C-terminus). Residues Leu123 to Ala132 and Ser335 to Lys336 from HER2-ECD were considered disordered and therefore not modelled. NM-02 forms a classical (variable-type) Ig-like fold, comprising of nine β-strands (Fig 3). The sequences of the three complementarity determining regions (CDR) according to the IMGT numbering system [27] are: CDR1 (Gly27-Asp38), CDR2 (Ile56-Thr65) and CDR3 (Ala105-Trp118). Two disulphide bonds (Cys23–Cys104 and Cys50–Cys112) are present. The second disulphide bond (Cys50–Cys112) covalently links framework 2 to CDR3, which itself exhibits an α-helical structure between residues Thr112A–Trp118 (IMGT numbering) [27].

HER2-ECD comprises a four-domain architecture, as previously reported [28]. Domains I-III are very similar to all published HER2 structures (max. Cα RMSD = 1.299 Å). Domain IV (residues 541–646) generally gave poor electron density and is therefore assumed to be disordered. Inspection of protein quaternary structure, reveals that NM-02 is bound to domains I and II of HER2-ECD. Assuming a classic topology, whereby domain IV is membrane proximal and precedes the transmembrane domain [29], this positions NM-02 within a potentially more accessible, membrane distal epitope (Fig 3a). Notably, this is distinct from the binding sites for the anti-HER2 therapeutic antibodies Pertuzumab [30] and Trastuzumab [31], which

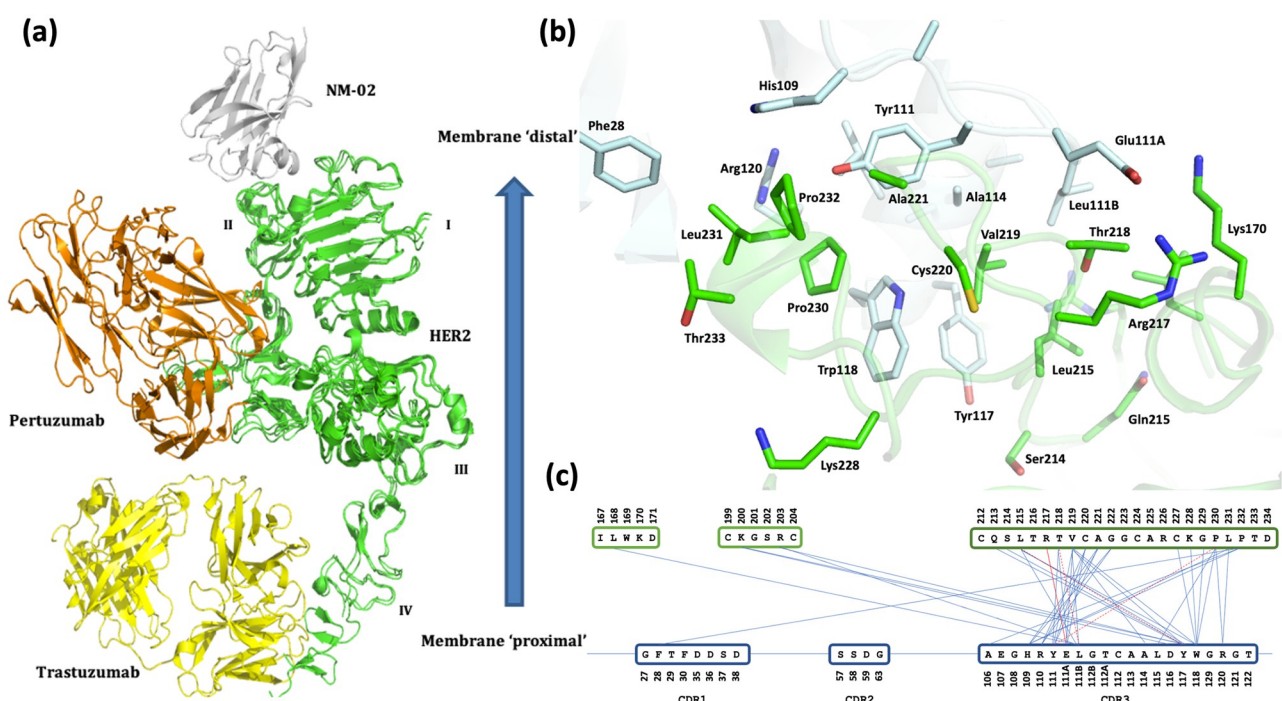

**Fig 3. Co-crystal structure of NM-02 in complex with HER2-ECD.** (a) Composite figure of NM-02 (grey) in complex with HER2-ECD (green) overlaid with co-crystal structures comprising Trastuzumab (yellow; PDB ID 1N8Z) and Pertuzumab (orange; PDB ID 1S78). (b) Stick view of interface between NM-02 (grey) and HER2-ECD (green). (c) Residue contact map, highlighting the number of interactions made by CDR3. Structural figures rendered in PyMOL.

corroborates both the applied selection criteria and obtained biochemical data (Fig 1). In addition, the HER2 epitope with NM-02 is different to that previously reported for an anti-HER2 camelid-derived sdAb [28].

## Features of the structural interface between NM-02 and HER2-ECD

A total of 16 residues from NM-02 comprise the paratope (defined as < 5 Å from HER2-ECD), with 12 amino acids relinquishing at least 10 Å$^2$ of solvent accessibility. It is predominantly defined by residues from CDR3, with minor contributions from CDR1 and framework1 (Fig 3b and 3c). The paratope comprises a mixture of both polar and hydrophobic contacts, including the participation of four aromatic (Phe28, Tyr111, Tyr117 and Trp118) and three aliphatic side chains (Leu4, Leu111B and Leu115).

A total of 21 residues from domains I and II of HER2-ECD, comprise the epitope (defined as < 5 Å from NM-02) and is composed entirely of loop regions (Fig 3). Notably, it includes several basic residues (Lys170, Lys200, Arg203 and Arg217), which gives the binding site on HER2-ECD a partial positive charge. The complimentary surface of NM-02 includes a number of acidic residues (including Glu111A and Asp115), with the carboxylate sidechain of Glu111A forming a salt bridge with the guanidino moiety of Arg217 from HER2-ECD. In total there are three hydrogen bonds and one salt bridge between HER2-ECD and NM-02 that contributes to both the specificity and stability of the protein complex formed.

Overall, NM-02 buries 637 Å$^2$ of surface area from bulk solvent upon complex formation and this is consistent with a mean surface area of 768.5 ± 201.0 Å$^2$, as determined for this class of antibody [32]. HER2-ECD buries a comparable surface area of 627 Å$^2$.

## Single-domain antibody engineering: Humanisation of NM-02

Antibody humanisation is a well-established approach for de-risking any potential immunogenicity issues, that can arise as a result of utilising animal-derived variable (V) domains as the basis for human therapeutics (for review of techniques *see* Strohl and Strohl [18]). However, far less information is available with respect to strategies for humanising sdAbs. Therefore, a surface-editing approach was adopted, based upon methodology proposed by Vincke et al. [33] to design fourteen humanised variants (A to N) derived from the parent NM-02 (S2 Table). These ranged from 77.3% to 84.5% human identity, as defined by comparison to nearest human germline V-gene segment [34]. The primary objective of the design process was to increase the number of human-equivalent residues, whilst still retaining wild type-like affinity, defined by us as being within 1.5-fold of that exhibited by NM-02.

Although the incorporation of human-equivalent residues into the pseudo-VH/VL interface (positions 49 and 50) of VHHs has previously been shown to be tolerable [33], this was not implemented for NM-02 as this would have introduced either a therapeutically undesirable deamidation motif and/or disrupted the second archetypal disulphide bond, respectively (S2 Table). Instead, additional human-equivalent residues were inserted into framework 1 and 3 in order to further increase human content. The overall strategy was therefore to design an initial set of variants (A–F) reaching desired maximal humanised content, before additional parental back mutations were incorporated (G–N), with the sole purpose of retaining wild type-like affinity for HER2-ECD (S2 Table).

## Kinetics of NM-02 humanised variants binding to HER2-ECD

The kinetics of the interaction between NM-02 humanised variants and the ECD of HER2 was examined using SPR (Fig 4). Obtained dissociation constant (KD) for NM-02 and wild type-like 'framework-swapped' controls were similar, as expected, and determined to be ~ 200 pM

## (a) Sequence Identity

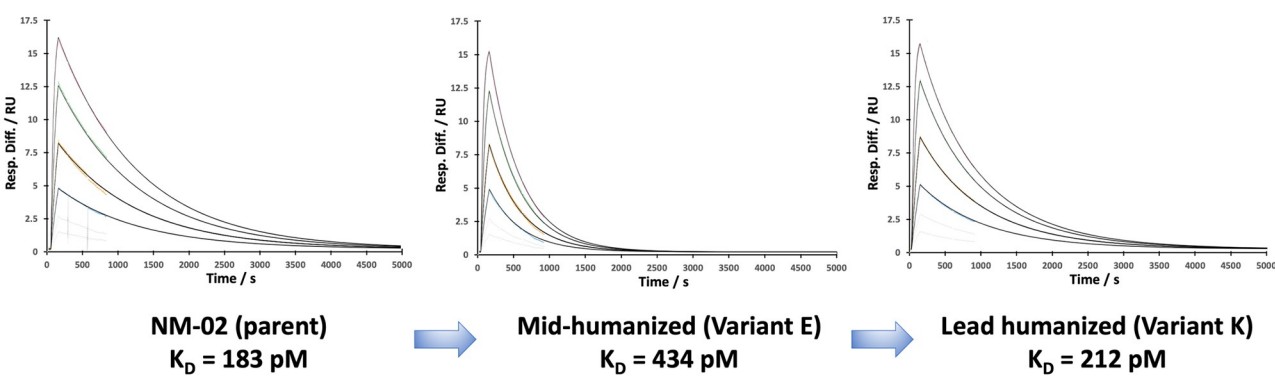

## (b) Surface Plasma Resonance (SPR)

**Fig 4.** Sequence and kinetic properties of NM-02 versus selected humanised candidates (a) Sequence alignment and human identity of NM-02 (parent), a mid-humanised candidate (variant E) and lead humanised candidate (variant K). Human equivalent substitutions into NM-02 are highlighted (grey) and a comparison to human germline IGHV3-64 is also shown. (b) Corresponding SPR sensorgrams, with the selected humanised lead (variant K) exhibiting near-parent-like affinity for HER2-ECD with elevated human content (82.5%). For sequence alignment, human identity and SPR sensorgrams of all variants in this study, see S2–S5 Tables.

(S2 Table). In contrast to that typically observed for a monoclonal antibody, NM-02 exhibits a general 'fast-on' and 'fast-off' kinetic profile, which is maintained by all humanised variants. Humanised variant K exhibits a near-equivalent dissociation constant (KD = 212 pM) to that of NM-02, while crucially maintaining an elevated human content (82.5%). Therefore, variant K was designated as the humanised lead candidate for follow-up developability profiling.

### Developability profiling of lead humanised NM-02 candidate

Biophysical characterisation of early-stage biologics is important to identify any associated risks or safety concerns, prior to beginning 'good manufacturing practice' (GMP) scale production and subsequently clinical trials [35]. This is often referred to as 'developability' profiling. Consequently, the lead humanised NM-02 candidate (variant K) was subjected to thermal, physical and enzymatic stress-tests, as well as solubility assessment to ensure good biophysical attributes were displayed.

With respect to thermal profiling, the lead humanised NM-02 candidate (variant K) exhibited improved thermal stability (Tm = 71.6˚C) versus the NM-02 parent (Tm = 67.8˚C) (Fig 5a). Interestingly, neither molecule exhibited detectable aggregation post-complete denaturation, as confirmed by simultaneous dynamic light scattering (DLS) measurements. This may be attributed to the additional disulphide bond, which in some cases has been recognised to improve stability and reduce aggregation propensity [36].

Freeze-thaw processes are common in drug manufacturing, as sample freezing provides a means to store product, prevent microbial growth and adjust in real-time to supply and demand [37]. To identify any potential liabilities associated with such stress, the lead

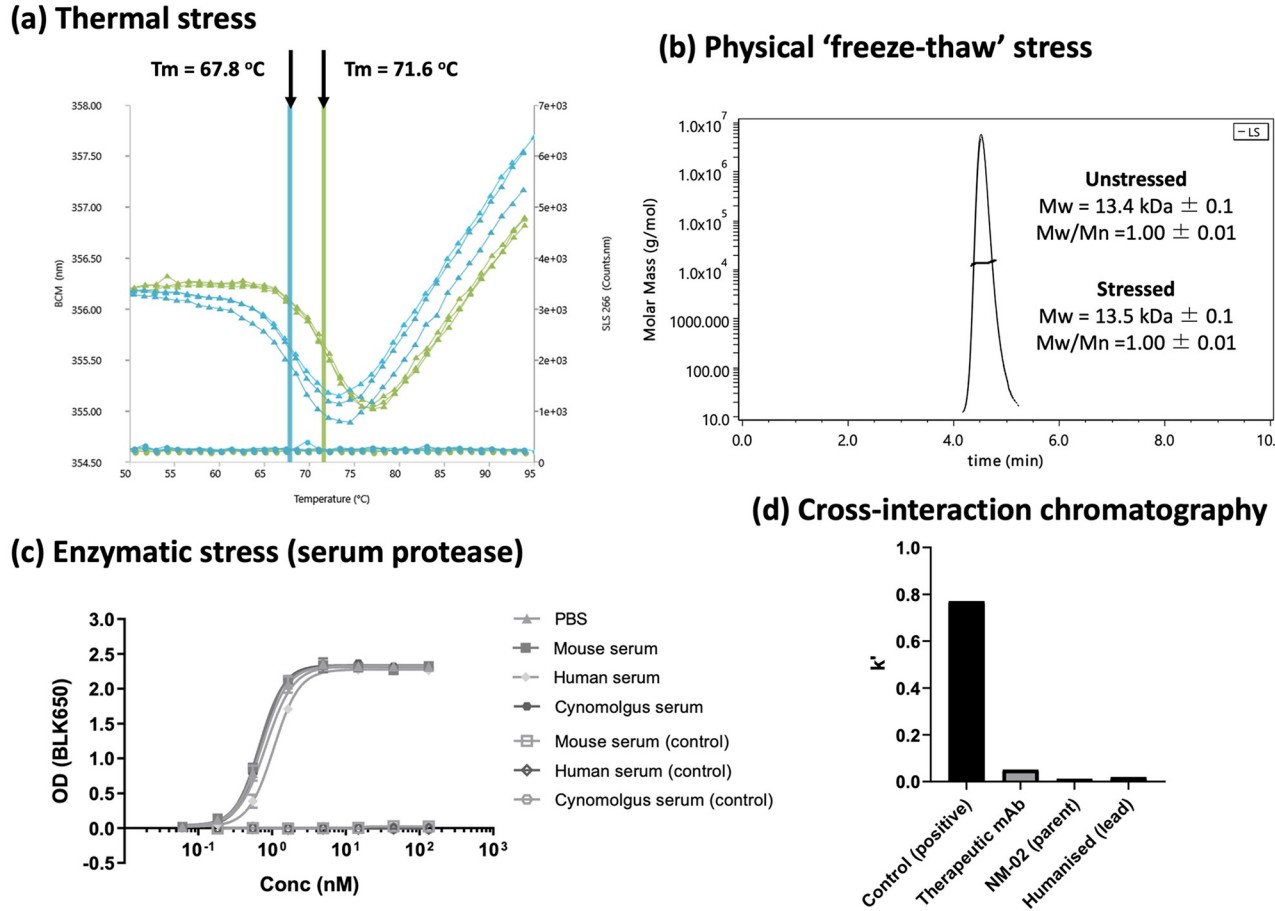

**Fig 5. Biophysical characterisation of lead humanised NM-02 candidate (variant K).** (a) Simultaneous intrinsic fluorescence (triangle) and static light-scattering (circle) measurements versus a thermal ramp for NM-02 (blue) and humanised lead (green). Observed melting temperature ($T_m$) highlighted with droplines and arrowhead. (b) Overlay of SEC-MALS profiles for lead humanised variant before (unstressed) and after (stressed) 10 cycles of freeze-thawing. (c) Potency of humanised lead variant following 7-day thermal incubation (37˚C) in mammalian sera of clinical relevance versus PBS. (d) Non-specific protein-protien interaction assessment using cross-interaction chromatography (CIC). Observed scores (k') are consistent (< 0.2) with that expected for a therapeutic mAb.

humanised NM-02 candidate (variant K) was subjected to ten cycles of 'freeze-thawing' and subsequently analysed for low-level aggregation by analytical size exclusion chromatography (SEC) coupled to a multi-angle light scattering (MALS) detector. This permitted for additional sensitivity of large aggregates, below that typically detected using UV alone. No significant increase in aggregation content was observed for the lead humanised NM-02 candidate (variant K) versus the unstressed sample (Fig 5b). The homogeneity of the sample across the main peak remained constant (polydispersity < 0.05) suggesting that despite using a simple phosphate buffered saline, pH 7.4 (PBS) formulation, the humanised variant exhibited a good developability profile with respect to freeze-thaw processes.

The stability of the lead humanised NM-02 candidate (variant K) in sera of clinical relevance, was assessed by incubating samples in mouse, cynomolgus monkey and human serum. After 7 days (37˚C), activity to recombinant HER2-ECD was maintained in all sera (Fig 5c) with only a minimal drop-off (1.5-fold) detected in human serum. This implies generally good resistance to serum proteases, which could function to degrade and reduce efficacy within a clinical setting.

To profile the propensity for non-specific protein-protein interactions, the lead humanised NM-02 candidate (variant K) was subject to analytical cross-interaction chromatography (CIC). Here, samples were passed across a column pre-loaded with polyclonal immunoglobulin (Ig) and retention time (s) determined (versus an unloaded column) to derive a score ($k'$). Scores below 0.2 typically correlate to a solubility > 100 mg/mL for mAbs [38]. The lead humanised NM-02 candidate (variant K) exhibited pass criteria similar to that of parent NM-02 and in line with that observed for a control therapeutic mAb (Fig 5d).

## SPECT/CT imaging of lead humanised NM-02 candidate in HER2+/BT474 xenograft mice

To assess the suitability of the lead humanised NM-02 candidate (variant K) for imaging purposes and therefore potential theranostic application, an *in vivo* biodistribution analysis was performed using orthotopic HER2+ breast cancer BT474 xenograft mice.

Single-domain antibodies were labelled with technetium-99m ($^{99m}$Tc) and injected intravenously (IV) to assess both image quality and tumour-target specificity. Using single-photon emission computed tomography/computed tomography (SPECT/CT), in combination with an X-ray scanner, specific accumulation of wild type NM-02 or lead humanised NM-02 candidate (variant K) was observed in tumours (5.2 ± 0.7% injected dose per gram (ID/g) and 5.2 ± 1.4% ID/g, respectively) (Fig 6). A moderate uptake of was observed in the liver (2.1 ± 0.2% ID/g for wild type NM-02 and 2.0 ± 0.1% ID/g for the lead humanised NM-02 candidate (variant K)), however this could be attributable to low levels of impurities in the radiolabeled preparations. Uptake in all other studied tissues remained below 1% ID/g, providing a high level of tumor conspicuity, as early as 90 minutes post-injection. Tumour specificity was further confirmed by cross-competing or 'blocking' of the radiolabelled response, using unlabelled sdAb, co-injected alongside (Fig 6). The overall *in vivo* performance of the lead humanised NM-02 candidate (variant K) was comparable to that of the parent (NM-02) and was, therefore, deemed to have retained the associated desirable properties, such as specificity and signal clarity, despite having undergone significant protein engineering.

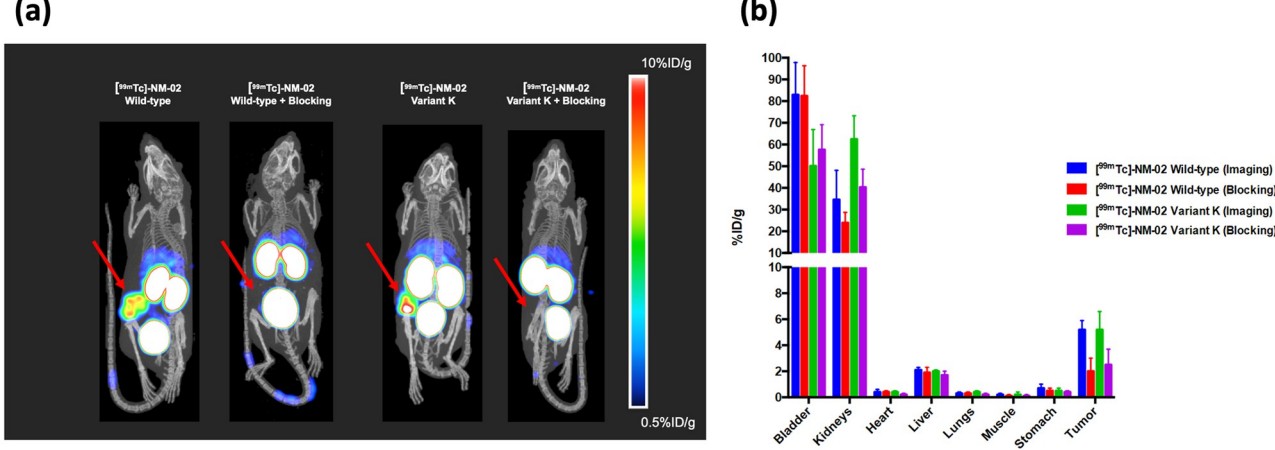

**Fig 6. SPECT/CT imaging of humanised anti-HER2 sdAb in HER2+ breast cancer BT474 xenograft model.** (a) Representative SPECT/CT images of HER2+ BT474 xenograft mice injected with either [$^{99m}$Tc]-NM-02 or [$^{99m}$Tc] humanised NM-02 (variant K) 90-mins post injection. In both cases, this signal can be specifically 'blocked' by co-injection of excess unlabelled sdAb. Red arrows denote location of subcutaneous HER2 positive BT474 xenograft. (b) Histogram depicting signal intensity on a per organ basis. Specific labelling of the xenograft tumor is clearly detected above baseline (as determined by non-targeted/non-associated organs). Strong signals arising from the urinary tract (bladder and kidneys) are consistent with renal clearance, as expected for small (< 69 kDa) biologics.

## Discussion

This study describes the discovery, biochemical characterisation, off-target screening, structural determination, humanisation, developability profiling and *in vivo* efficacy of NM-02, in order to demonstrate its tractability as a potential theranostic tool. The sdAb is highly-specific for HER2-ECD, recognising a distinct membrane-distal epitope, which is non-overlapping with current front-line therapeutics. Hence, combinatorial approaches including NM-02 could be further explored, especially where Trastuzumab resistance is a recognised issue [39]. Regulatory precedence for the humanisation of sdAbs has been set by Calplacizumab (Cablivi), the first FDA-approved sdAb therapy available [40]. Unlike some other humanised sdAbs, we have not used a pre-humanised scaffold as an acceptor for CDR-grafting [41]. Instead, a bespoke humanisation approach was used which potentially maximises the retention of wild type properties (*e.g.* affinity), whilst achieving increased human content. As observed in the kinetic data, this is frequently a trade-off, as the most human-like candidate is not necessarily the most potent. Hence, variant K was chosen as the lead humanised candidate, being at least two-fold more potent than variant E with only a slight reduction in human content (82.5% and 83.5% human identity, respectively).

Extensive stress-testing and developability profiling reveals desirable antibody drug-like properties, which de-risks the lead humanised NM-02 candidate (variant K) for progression along the clinical stage pipeline. Such properties appear to be also present in the wild type molecule, which implies more generally that these have been retained rather than 'engineered in'. From a candidate selection process, therefore early stage discovery screening may be beneficial to downstream developability success rates, as is well recognised for monoclonal antibodies [42].

Good SPECT/CT image quality of the lead humanised NM-02 candidate (variant K) has further been retained, as a result of specific tumour accumulation ($> 5\%$ ID/g). With respect to the biodistribution study, both wild type NM-02 and the lead humanised candidate presented similarly, characteristic of compounds of this size: rapid renal extraction followed by urinary excretion. Consequently, high levels of radioactivity, exceeding 30% ID/g, were observed in the kidneys and urinary bladder, as expected.

Taken all together, the dataset presented here demonstrates that NM-02 (and the corresponding lead humanised candidate) are highly specific and robust anti-HER2 sdAbs with good 'drug-like' properties. Comprising full epitope characterisation, sub-nanomolar potency, favourable biophysical properties and evidenced *in vivo* efficacy, either would be suitable for further clinical development as a potential anti-HER2 theranostic tool.

## Supporting information

**S1 Table. X-ray data and refinement statistics.**
(TIF)

**S2 Table. Humanised variants of NM-02.** The primary sequence for NM-02 was numbered according to IMGT [16] and inspected against a published template, human germline(s) and the experimentally-determined crystal structure (*see* Fig 3). Fourteen humanised variants (A to N) were designed for this study. Human germline residues incorporated into each NM-02 variant are coloured blue (common to both IGHV3-23 and IGHV3-64), pink (unique to IGHV3-23) or green (unique to IGHV3-64) respectively. A single non-germline residue (position 123; framework 4) was also examined (yellow). The corresponding human identity (%) of each NM-02 humanised variant is also shown.
(TIF)

**S3 Table. SPR sensorgrams for NM-02 (parent) and humanised variants (B, D and E).**
(TIF)

**S4 Table. SPR sensorgrams for NM-02 humanised variants (E-I).**
(TIF)

**S5 Table. SPR sensorgrams for NM-02 humanised variants (J-N).**
(TIF)

## Acknowledgments

The authors would like to acknowledge the efforts and contribution of our service contract partners at Retrogenix and Invicro, as well as the radiochemistry and biology groups at Queen Mary University of London (QMUL).

## Author Contributions

**Conceptualization:** Kovilen Sawmynaden, Gareth Hall.

**Investigation:** Kovilen Sawmynaden, Nicholas Wong, Sarah Davies, Richard Cowan, Richard Brown, David Tang, Maud Henry, Gareth Hall.

**Supervision:** Kovilen Sawmynaden, David Tickle, David Matthews, Mark Carr, Preeti Bakrania, Hong Hoi Ting, Gareth Hall.

**Writing – original draft:** Kovilen Sawmynaden, Gareth Hall.

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
