## [Decision Letter · Decision Letter 0]

28 Apr 2023

PONE-D-23-03760Co-crystallisation and humanisation of an anti-HER2 single-domain antibody as a theranostic toolPLOS ONE

Dear Dr. Hall, Thank you for submitting your manuscript to PLOS ONE. After careful consideration, we feel that it has merit but does not fully meet PLOS ONE’s publication criteria as it currently stands. Therefore, we invite you to submit a revised version of the manuscript that addresses the points raised during the review process.We would like you to address the specific changes as requested by reviewer 1 with additional inputs from reviewer 2 where deemed necessary.

We look forward to receiving your revised manuscript.

Kind regards,

Yolandy Lemmer

Academic Editor

PLOS ONE

Journal Requirements:

2. To comply with PLOS ONE submissions requirements, in your Methods section, please provide additional information regarding the experiments involving animals and ensure you have included details on (1) methods of sacrifice, (2) efforts to alleviate suffering.

“I confirm that I have mentioned all organizations that funded my research in the

Acknowledgements section of my submission, including grant numbers where

appropriate.”

“LifeArc provided humanisation service(s) to NanoMab under contractual arrangement. LifeArc is a self funded medical charity (registered with the Charity Commission for England and Wales no. 1015243 and a charity registered in Scotland with the Office of the Scottish Charity Regulator no. SC037861).”

“I confirm that I have mentioned all organizations that funded my research in the

Acknowledgements section of my submission, including grant numbers where

appropriate.”

“The authors declare the following financial interests/personal relationships which may be considered as potential competing interests: LifeArc provided the antibody humanisation service(s) under contractual arrangement with NanoMab”

6. Please amend your authorship list in your manuscript file to include author David Tickle.

7. Please include a separate caption for each figure in your manuscript.

Reviewers' comments:

Reviewer's Responses to Questions

**Comments to the Author**

1. Is the manuscript technically sound, and do the data support the conclusions?

Reviewer #1: Yes

Reviewer #2: Yes

2. Has the statistical analysis been performed appropriately and rigorously? 

Reviewer #1: Yes

Reviewer #2: Yes

3. Have the authors made all data underlying the findings in their manuscript fully available?

Reviewer #1: Yes

Reviewer #2: Yes

4. Is the manuscript presented in an intelligible fashion and written in standard English?

Reviewer #1: No

Reviewer #2: Yes

5. Review Comments to the Author

Reviewer #1: I read the article submitted by Sawmynaden et al with great interest. It describes the identification, humanisation, complex structure determination, and characterization of an anti-HER2 single-domain antibody and its target human HER2.

Technically and in terms of interpretation, the article is well written and requires no further modification. My only concern is with the narrow, jargonistic linguistic style - with occasional strangely dated conjunctions. While this style may be suitable for an inner circle of aficionados, I suggest that the text could be made more accessible by replacing complex phrases with more generally comprehensible ones.

Thus, "frontline treatment options" could be replaced with "current best competing antibodies," "de-risk" with "avoid immune response," "progression towards" with "proceed to" or "allow for," "chemistry, manufacturing, and control" with "pharmaceutical quality," "performed full developability profiling" with "assessed the pharmacological suitability," and "applicability" with "usefulness," among others.

I also suggest removing as many brackets as possible to improve readability. For instance, instead of "show a marked increase in expression levels (2 to 20-fold)," one could say "show a two- to twenty-fold improvement in expression."

On page 2, line 40, I recommend removing "As a result" or replacing it with "Correspondingly." There is no direct causal relationship between the sentences.

A careful checking of the text remaining should reveal similar minor adjustments that could render the text more generally comprehensible - and hence more citable.

Reviewer #2: The manuscript is well written and well presented. However, please allow me to make a few minor comments and suggestions. When reporting scientific data, it is recommended that you report in past tense and not in present tense or as absolutes; replace is with was/ were where relevant etc. Additionally, please avoid the use of pronouns e.g., we, our etc.

Please see attachment for a detailed review.

6. PLOS authors have the option to publish the peer review history of their article (what does this mean?). If published, this will include your full peer review and any attached files.

Reviewer #1: **Yes: **Wolf-Dieter Schubert

Reviewer #2: No

---

## [Author Response · Author response to Decision Letter 0]

20 Jun 2023

Dear Dr Lemmer

Please find enclosed a revised version of our paper “Co-crystallisation and humanisation of an anti-HER2 single domain antibody as a theranostic tool” (Manuscript Number PONE-D-23-03760). We were pleased to read that both reviewers assessed the work with great interest and that they felt the manuscript contained important information of interest to other investigators. The reviewers had highlighted several grammatical concerns that we feel we have now addressed in a revised manuscript. We have summarised the changes made to address the reviewers’ comments below and, as requested, we have provided a copy of the original manuscript with tracked changes to facilitate the review process. In addition, we have addressed the Journal requirements below, as highlighted in your last correspondence.

1. The manuscript has been adapted to comply with PLOS ONE’s style requirements, including changing font styles, reordering sections, referencing and preparing individual .TIFF files for the figures.

2. To comply with PLOS ONE’s submission requirements, in the methods section covering animal experiments we have included further details on (1) methods of sacrifice, (2) methods of anaesthesia and/or analgesia, and (3) efforts to alleviate suffering.

(1) “Mice under anaesthesia were sacrificed by cervical dislocation, after confirmation that all SPECT/CT images were technically acceptable.”

(2) “Mice were subcutaneously inoculated on the right flank with BT474 cells (6 x106 cells suspended in 1:1 (v/v) matrigel:PBS) following implantation of estrogen pellets, using buprenorphine analgesic.”

“Whole-body SPECT/CT images were acquired 1.5 hr post-injection in a VECTor6-CTXUHR (MILabs) preclinical imaging system under isoflurane anaesthesia.”

(3) “The animals were observed daily for any abnormalities indicative of health problems, to prevent suffering. Only healthy animals were placed on the study.” 

3. “I confirm that I have mentioned all organizations that funded my research in the Acknowledgements section of my submission, including grant numbers where appropriate.”

a) This work was funded by the research charity LifeArc, under contractual arrangement with NanoMab. LifeArc funded the research activity carried out by the University of Leicester group.

b) LifeArc and NanoMab were actively involved in the study design, data collection and analysis, as well the decision to publish and preparation of the manuscript. 

c) Kovilen Sawmynaden, Sarah Davies, Richard Brown, David Tang, Maud Henry, David Tickle, David Matthews and Preeti Bakrania received a salary from LifeArc. Nicholas Wong and Hong Hoi Ting received a salary from NanoMab.

4. For the Acknowledgments section:

“I confirm that I have mentioned all organizations that funded my research in the Acknowledgements section of my submission, including grant numbers where appropriate.”

5. For the Competing Interests section: 

“The authors declare the following financial interests/personal relationships which may be considered as potential competing interests: LifeArc provided the antibody humanisation service(s) under contractual arrangement with NanoMab. This does not alter our adherence to PLOS ONE policies on sharing data and materials.”

6. We have amended the authorship list to include David Tickle.

7. We have included a separate caption for each figure directly after the paragraph that first cites that figure, as highlighted in the style requirement. 

8. We have included captions for the Supporting Information at the end of the manuscript, as highlighted in the style requirement.

9. We are happy that the reference list is complete and correct. 

Reviewer 1

We would like to thank the reviewer for their comments. We were pleased to read that the reviewer considered the manuscript technically well-written, although they had they felt that the linguistic style contained complex and jargonistic phrases, and that these should be replaced with more generally comprehensive ones. We agree with this comment and, as such, we have adapted the manuscript accordingly. Along with the changes suggested by the reviewer, we have also rephrased several additional sentences to improve the accessibility of the manuscript. We have also reduced the number of brackets used throughout the manuscript to improve the flow and readability.

Reviewer 2

We would like to thank the reviewer for their comments. The reviewer felt that the paper was well written and well presented. We would also like to thank them for making minor suggestions regarding the use of pronouns and tenses. As a result, we have carefully ensured that the report is written in the past tense, and we have also significantly reduced the use of pronouns.

In addition to changes made in response to the reviewers’ comments, we have addressed all comments made within the ‘Revised Manuscript with Tracked Changes’ document, as well as correcting additional typographical and grammatical errors identified in the original paper. We hope that the revised paper fully addresses the concerns raised by the two reviewers and is now considered acceptable for publication in PLOS ONE.

Yours Sincerely,

Gareth Hall

---

## [Editor Report · Decision Letter 1]

22 Jun 2023

Co-crystallisation and humanisation of an anti-HER2 single-domain antibody as a theranostic tool

PONE-D-23-03760R1

Dear Dr. Hall,

We’re pleased to inform you that your manuscript has been judged scientifically suitable for publication and will be formally accepted for publication once it meets all outstanding technical requirements.

Kind regards,

Yolandy Lemmer

Academic Editor

PLOS ONE

---

## [Editor Report · Acceptance letter]

7 Jul 2023

PONE-D-23-03760R1 

Co-crystallisation and humanisation of an anti-HER2 single-domain antibody as a theranostic tool 

Dear Dr. Hall:

I'm pleased to inform you that your manuscript has been deemed suitable for publication in PLOS ONE. Congratulations! Your manuscript is now with our production department. 

Kind regards, 

on behalf of

Dr. Yolandy Lemmer 

Academic Editor

PLOS ONE